# Abnormal Dynamic Reconfiguration of Multilayer Temporal Networks in Patients with Bipolar Disorder

**DOI:** 10.3390/brainsci14090935

**Published:** 2024-09-19

**Authors:** Luyao Lai, Dandan Li, Yating Zhang, Jianchao Hao, Xuedong Wang, Xiaohong Cui, Jie Xiang, Bin Wang

**Affiliations:** College of Computer Science and Technology (College of Data Science), Taiyuan University of Technology, Taiyuan 030024, China; lailuyao0425@link.tyut.edu.cn (L.L.); zhangyating0747@link.tyut.edu.cn (Y.Z.); haojianchao0708@link.tyut.edu.cn (J.H.); wangxuedong0730@link.tyut.edu.cn (X.W.); cuixiaohong@tyut.edu.cn (X.C.); xiangjie@tyut.edu.cn (J.X.); wangbin01@tyut.edu.cn (B.W.)

**Keywords:** bipolar disorder, multilayer temporal network, phase coherence, inter-layer coupling, dynamic reconfiguration

## Abstract

Background: Multilayer networks have been used to identify abnormal dynamic reconfiguration in bipolar disorder (BD). However, these studies ignore the differences in information interactions between adjacent layers when constructing multilayer networks, and the analysis of dynamic reconfiguration is not comprehensive enough; Methods: Resting-state functional magnetic resonance imaging data were collected from 46 BD patients and 54 normal controls. A multilayer temporal network was constructed for each subject, and inter-layer coupling of different nodes was considered using network similarity. The promiscuity, recruitment, and integration coefficients were calculated to quantify the different dynamic reconfigurations between the two groups; Results: The global inter-layer coupling, recruitment, and integration coefficients were significantly lower in BD patients. These results were further observed in the attention network and the limbic/paralimbic and subcortical network, reflecting reduced temporal stability, intra- and inter-subnetwork communication abilities in BD patients. The whole-brain promiscuity was increased in BD patients. The same results were observed in the somatosensory/motor and auditory network, reflecting more functional interactions; Conclusions: This study discovered abnormal dynamic interactions of BD from the perspective of dynamic reconfiguration, which can help to understand the pathological mechanisms of BD.

## 1. Introduction

Bipolar disorder (BD) is a chronic, relapsing severe illness usually associated with abnormal executive and memory functions and cognitive deficits [1,2,3,4,5]. Based on resting-state functional magnetic resonance imaging (rs-fMRI) techniques, many studies have shown that cognitive impairment in BD patients is closely related to abnormal functional connectivity (FC) between brain regions [6,7,8]. Massalha et al. [9] found that BD patients have abnormal FC in the default mode network and the sensorimotor network, affecting executive functions. McPhilemy et al. [10] identified abnormal FC in spatial memory-related subnetworks in BD patients, indicating an inhibition of memory processing in the resting state. Hu et al. [11] found that FC between the dorsal and ventral attention networks was positively associated with cognitive deficits in BD patients. These studies indicate that abnormal FC in the resting-state are important markers of BD, which can help us better explore the cognitive mechanisms of BD. However, these studies assume that the brain is static and ignore the dynamic interactions that occur in the brain over time.

Multilayer temporal network is a novel network model. It contains both intra-layer connections and inter-layer connections [12]. The intra-layer connections can represent the interaction between brain regions at each time point, and the inter-layer connections capture changes in the FC of the brain over time. Multilayer brain networks have been widely used to quantitatively characterize the functional connectivity of the brain and to simulate the evolution of the network over time [13,14]. Compared to traditional static networks, multilayer brain networks have been shown to have advantages in modeling brain dynamics over time [15,16]. It can accurately capture the dynamic changes of the brain, displaying the order and relationship of interactions across time. Previous studies have identified anomalies in the resting-state FC of the BD, while the brain is dynamically changing over time. Therefore, we hypothesized that the abnormalities of intra-layer and inter-layer connections in BD patients could be further identified by constructing multilayer temporal networks.

Dynamic reconfiguration is a manifestation of changes in brain interactions and is increasingly being employed to tap into functional changes in brain disorders. The promiscuity, switching rate, and the recruitment and integration coefficients are commonly used metrics to quantify dynamic reconfiguration. The promiscuity and switching rate are quantified in the aspect of participation and frequency of changes. The recruitment coefficient and integration coefficient are quantified in terms of intra-subnetwork and inter-subnetwork module allegiance. Many studies have employed these metrics to explore the abnormal dynamic reconfiguration of common mental disorders such as major depression [17], schizophrenia [18], and post-traumatic stress disorder [19]. It is shown that dynamic reconfiguration provides insights into the dynamic changes in brain network interactions in psychiatric disorders. Recent studies have found that the brain networks of BD patients have reduced network switching rates of key hubs in the default mode network during dynamic reconfiguration [20]. However, this study only quantified the dynamic reconfiguration of BD patients from a single perspective. Meanwhile, it ignored the differences in information interactions between adjacent layers when constructing the network. Therefore, we hypothesized that it would be possible to explore the abnormal reconfiguration of BD patients from different perspectives by constructing more comprehensive multilayer temporal networks.

In this study, we constructed a multilayer temporal network for each subject based on rs-fMRI data from BD patients and normal control (NC) subjects. The intra-layer connections were constructed using the phase coherence method, and the inter-layer connections of the nodes were assessed over time using network similarity. Afterwards, we further explored the inter-layer coupling. Finally, the promiscuity, recruitment coefficient, and integration coefficient were calculated based on the segmented community. In conclusion, this study aims to explore the dynamic connectivity anomalies of brain networks and dynamic changes in community structure from another perspective.

## 2. Materials and Methods

### 2.1. Participants

This study utilized rs-fMRI data obtained from the OpenfMRI public database (https://www.openfmri.org/, accessed on 16 January 2023), which was made available by the University of California, Los Angeles (UCLA) Consortium for Neuropsychiatric LA5c study. All participants provided written informed consent following the procedures approved by the UCLA Institutional Review Board. The experiment comprised 46 patients diagnosed with BD (bipolar I) and 54 NC subjects. Diagnoses followed the Diagnostic and Statistical Manual of Mental Disorders and were based on the Structured Clinical Interview for DSM-IV (SCID-I). The inclusion criteria for subjects were males or females between the ages of 21 and 50 years who had completed at least 8 years of formal education and spoke English or Spanish as their primary language. Moreover, subjects who were left-handed, pregnant, or had other contraindications to scanning were excluded. The specific demographic information is presented in Table 1.

### 2.2. Imaging Acquisition and Preprocessing

The neuroimaging data in this study were acquired using a 3T Siemens Trio scanner. Participants were instructed to remain relaxed with open eyes and not receive any stimuli or engage in specific thoughts during the acquisition. The imaging parameters were set as follows: slice thickness = 4 mm, slices = 34, repetition time = 2 s, echo time = 30 ms, flip angle = 90°, field of view = 192 mm, and matrix = 64 × 64.

The Data Processing and Analysis for Brain Imaging (DPABI, http://rfmri.org/dpabia, accessed on 17 January 2023) toolbox was utilized for preprocessing the data. The first ten volumes were discarded to mitigate the initial effects of scanning. Slice-timing and head-motion correction were applied, with subjects excluded if head-motion exceeded 2 mm or rotation angle exceeded 2°. The resulting data were then transformed into standard Montreal Neurological Institute space, spatially smoothed using 6-mm-full-width at half maximum Gaussian kernel convolution, band-pass filtered with 0.01 ≤ f ≤ 0.1 Hz to remove covariates, and a functionally defined atlas was used to divide the resting-state scans into 116 regions of interest (ROIs). Time series were extracted for the 90 brain regions (nodes) remaining after removal of the cerebellum.

Based on the Anatomical Automatic Labeling (AAL) atlas [21], the 90 ROIs were functionally parcellated into five distinct resting-state functional networks (RSNs). The identified RSNs included the somatosensory/motor and auditory (MAN), visual (VN), attention (AN), default mode (DMN), and limbic/paralimbic and subcortical network (LSN). Information on the mapping of brain regions and functional subnetworks is provided in Appendix A.

### 2.3. Multilayer Temporal Network Construction

The time series extracted from the 90 ROIs were analyzed at each time point, and then intra-layer connections and inter-layer couplings were established to construct the multilayer temporal network.

#### 2.3.1. Intra-Layer Connections Construction Based on Phase Coherence

To capture the FC of the brain at each time point, we constructed intra-layer connections for each subject based on the phase coherence method [22,23,24,25]. Phase coherence is an instantaneous measurement with maximum temporal resolution [26], enabling more accurate estimates of time-varying FC (TVFC). The phase coherence method does not require a window length to be set, overcoming the limitations of the sliding time window correlation method [27] where the window length is difficult to determine [25].

To calculate the instantaneous phase of brain region n at each time point, the Hilbert transform was initially applied to the BOLD signal. Subsequently, the BOLD phase coherence between brain regions n and p at time t for each subject was determined using the following equation:(1)$$TVFC\left(n,p,t\right)=cos\left(\theta \left(n,t\right)-\theta \left(p,t\right)\right)$$

The range of $TVFC\left(n,p,t\right)\;$ is from −1 to 1. The closer $TVFC\left(n,p,t\right)\;$ is to 1, the more synchronized the BOLD signals of brain regions n and p are at time t. Conversely, if the BOLD signals exhibit orthogonality (with a phase difference of 90°), $TVFC\left(n,p,t\right)\;$ approaches 0. Considering the unclear biological explanation for negative correlation [28,29], we only regard positive correlation in this study and set negative correlation values as zero.

#### 2.3.2. Inter-Layer Connections Construction Based on Network Similarity

To describe the dynamic characteristics of the brain network more completely, it is also necessary to construct connections between adjacent temporal layers. Considering that using the same parameters to represent the inter-layer connectivity between different temporal layers would ignore the potential differences between different nodes [30], we use the network inter-layer similarity to calculate the inter-layer coupling strength ω of each node with the following formula:(2)$$\omega_{i}^{\left(t,t+1\right)}=\frac{\sum_{j}a_{ij}\left(t\right)a_{ij}\left(t+1\right)}{\sqrt{\left[\sum_{j}a_{ij}\left(t\right)\right]\left[\sum_{j}a_{ij}\left(t+1\right)\right]}}$$
where $a_{ij}\left(t\right)\;$ and $a_{ij}\left(t+1\right)$ are the elements in the adjacency matrix of the adjacent temporal network, $a_{ij}\left(t\right)\;$ = 1 if the node i is connected to the node j in the temporal layer t, otherwise $a_{ij}\left(t\right)\;$ = 0. A higher value of $\omega_{i}$ indicates that node i exhibits more remarkable similarity at adjacent time points.

### 2.4. Multilayer Community Detection

Based on the constructed multilayer temporal network, we use the GenLouvain method [31] to identify communities over time. The algorithm obtains information about the division of nodes into communities by maximizing a multilayer modular quality function Q. Moreover, we further consider the differences in temporal coupling of different nodes in adjacent temporal layers when calculating Q, which is calculated as follows:(3)$$Q=\frac{1}{2\mu }\underset{ijlr}{\sum }\left[\left(A_{ijl}-\gamma_{l}\frac{k_{il}k_{jl}}{2m_{l}}\right)\delta_{lr}+\delta_{ij}\omega_{jlr}\right]\delta \left(g_{il},g_{jr}\right)$$
where μ  is the total weight of edges in the multilayer network; $A_{ijl}$ represents the functional connection strength between node i and node j in layer *l*; $k_{il}k_{jl}/2m_{l}$ represents the correlation matrix in the Newman-Girvan null model, where $k_{il}\;$ and $k_{jl}$ are the total weights of all edges connected to node i and node j, and $\;m_{l}\;$ is the total weighting degree of all nodes in layer *l*; $\;\delta_{lr}\;$ is the Kronecker delta function, when l = r, $\delta_{lr}=1\;$ and 0 otherwise [13].$\;g_{il}\;$ and $g_{jr}\;$ represent the communities to which node i belongs in layer l and node j belongs in layer r, respectively. $\gamma_{l}$ refers to the structural resolution parameter; $\;\omega_{jlr}\;$ denotes the inter-layer coupling parameter of node *j* between layers *l* and *r*. In this experiment, γ is set to 1, and the value of ω is calculated using Equation (2).

### 2.5. Dynamic Reconfiguration Metrics

For each subject, we calculated promiscuity, recruitment, and integration coefficients to characterize the time-varying properties of the multilayer network from different perspectives.

The promiscuity metric quantifies the proportion of communities in which a node participates among all communities [32]. The calculation formula for the promiscuity coefficient of node i is as follows:(4)$$P_{i}=\frac{m_{i}-1}{M-1}$$
where $m_{i}$ is the number of communities in which the node is involved, and M is the total number of communities. P=0 indicates minimum involvement (participation in only one community), and P=1 indicates maximum involvement (participation in all communities).

The recruitment coefficient quantifies the average intra-network module allegiance of a node [33] and indicates the likelihood of a node belonging to the same community as the nodes within its functional network. The recruitment coefficient for node i in functional network S can be calculated using the following formula:(5)$$R_{i}^{S}=\frac{1}{n_{S}}\underset{j\in S}{\sum }MA_{ij}$$
where $n_{S}$ is the number of nodes in the functional network S. $MA_{ij}$ is the probability that node i and node j belong to the same community. If in the same community, $MA_{ij}=1$, otherwise 0 and averaged over all time layers and all runs to get the result [34].

Similar to recruitment, the integration coefficient quantifies the average inter-network module allegiance of a node [35], which represents the probability that the node belongs to the same community as a node in another functional network. The integration coefficient for node i in the functional network S can be calculated as follows:(6)$$I_{i}^{S}=\frac{1}{N-n_{S}}\underset{j\notin S}{\sum }MA_{ij}$$
where N represents the number of nodes.

Due to the inherent uncertainty in the results of community partitioning [36,37], we repeated the community detection algorithm 100 times and calculated average dynamic reconfiguration metrics (promiscuity, recruitment, and integration coefficients) based on the results of these 100 runs. The overall flowchart is shown in Figure 1.

### 2.6. Statistical Analysis

To identify the existence of significant group differences in age and education, separate two-tailed *t*-tests were conducted for these variables. Meanwhile, gender data were analyzed by χ^2^ test. The independent-samples *t*-test was employed to detect between-group differences in multilayer network measures. A false discovery rate (FDR) correction was applied to account for multiple comparisons, with a critical value of *p* < 0.05 indicating statistical significance. Furthermore, Spearman’s correlation analyses were conducted to examine the associations between dynamic reconfiguration metrics and symptom severity in the BD groups.

## 3. Results

### 3.1. Group Differences at Whole-Brain Level

Metrics at the whole-brain level were obtained by averaging the values for each brain region across all subjects. Specifically, the promiscuity was significantly increased in BD patients compared to controls (t = 2.308, *p* = 0.023, Figure 2B). The inter-layer coupling strength (t = −2.324, *p* = 0.022, Figure 2A), recruitment coefficient (t = −2.968, *p* = 0.004, Figure 2C), and integration coefficient (t = −2.292, *p* = 0.024, Figure 2D) were significantly lower in BD patients.

### 3.2. Group Differences at Subnetwork Level

At the subnetwork level, BD patients had significantly lower inter-layer coupling strength (t = −3.259, corrected *p* = 0.008, Figure 3A) in the LSN. The promiscuity was significantly increased in MAN (t = 2.209, corrected *p* = 0.049, Figure 3B). The recruitment coefficients of BD patients were significantly lower in the MAN (t = −2.261, corrected *p* = 0.043), AN (t = −2.726, corrected *p* = 0.038) and DMN (t = −2.513, corrected *p* = 0.034) (Figure 3C). The integration coefficient of BD patients in LSN (t = −3.076, corrected *p*= 0.014, Figure 3D) was significantly lower than the NC group. The *p*-values before correction are shown in Appendix A.

To further identify in which group of subnetwork pairs the integration coefficients of BD patients were significantly different, we further explored the integration coefficients between subnetworks. The results showed that BD patients had significantly lower integration coefficients than controls between the two subnetwork pairs, including the AN-LSN (t = −3.055, corrected *p* = 0.029, Figure 4A) and DMN-LSN (t = −2.892, corrected *p* = 0.024, Figure 4B).

### 3.3. Group Differences at Node Level

At the nodal level, compared with the NC group, BD patients had significantly lower inter-layer coupling in two brain regions, one of which was located in the MAN and the other in the LSN (Figure 5A). The recruitment coefficient of BD patients was significantly lower in five brain regions (Figure 5B), three of which belonged to the AN, and the other two belonged to the LSN and the VN. The integration coefficients were significantly lower in eleven brain regions (Figure 5C), eight of which belonged to LSN, two to DMN, and one to AN. The promiscuity was not significantly different between the two groups. The *p*-values before and after correction are shown in Appendix A.

### 3.4. Correlations between Dynamic Reconfiguration Measures and Clinical Measurements

In our study, the correlation between the dynamic reconfiguration metrics and the HAMD and YMRS scores was investigated using Spearman’s correlation for areas where there were statistically significant differences between groups. The results showed that the integration coefficient in the whole brain was significantly and positively correlated with the HAMD (r = 0.336, *p* = 0.022, Figure 6A) and YMRS (r = 0.339, *p* = 0.021, Figure 6B) scores. At the nodal level, the integration coefficient of the right temporal gyrus of the temporal pole (TPOmid) region was also significantly positively correlated with HAMD (r = 0.333, *p* = 0.024, Figure 6C) and YMRS (r = 0.298, *p* = 0.044, Figure 6D) scores.

## 4. Discussion

In this study, the multilayer temporal network model was used to explore the abnormal reconfiguration of the dynamic network of BD patients. The network is modeled across time by constructing intra-layer connections and inter-layer coupling. The inter-layer coupling differences between nodes are further explored afterward. Finally, we quantified dynamic modular reconfiguration using promiscuity, recruitment, and integration coefficients. We found increased promiscuity in the BD group compared to the NC group. The inter-layer coupling strength, recruitment, and integration coefficients were significantly lower. In addition, these changes correlated with individual symptom severity. Thus, our study links dynamic changes in BD patients with clinical outcomes, providing new perspectives to explain the pathophysiological mechanisms of the disease.

### 4.1. Reduced Inter-Layer Coupling of BD Is Associated with Brain Emotion Regulation Function

In our study, the global inter-layer coupling strength was reduced in BD patients. Inter-layer coupling is an assessment of the similarity of neighboring temporal layers, which can reflect temporal stability. As such, the decrease in inter-layer coupling in BD patients implies the decrease in temporal stability. We further found that the reduction of global stability may be mainly caused by the LSN system. Therefore, we speculate that the LSN system plays an important role in BD patients. Previous studies have found increased FC variability in the LSN in BD, reflecting reduced temporal stability [38]. Impairment of the LSN has been shown to have a significant impact on with BD [39], which is consistent with our findings. In addition, several brain regions of the LSN system are over-activated during emotional processing and regulation [40].

The results of our work showed that the reduced inter-layer coupling is more indicative of an instability in the LSN of BD patients, which may lead to impaired emotional processing and regulation. Meanwhile, we found that the inter-layer coupling strength was significantly reduced in the TPOmid.R region, which also belongs to LSN. It further confirmed that the reduced temporal stability in BD patients may be caused by LSN. In summary, the changes in inter-layer coupling in the LSN may influence the emotion regulation function of the BD.

### 4.2. Abnormal Promiscuity of BD Is Associated with Brain Sensorimotor Function

In our study, whole-brain promiscuity was significantly increased in BD patients. The promiscuity describes the extent to which nodes are involved in different brain functional modules [41]. It measures the community participation of brain regions and can reflect the functional interaction of the brain regions. Increased promiscuity represents that BD patients are involved in more functional modules, reflecting more functional interactions. In addition, we found that BD patients had significantly increased promiscuity in the MAN system. Our results indicates that there are more functional interactions between MAN system and subnetworks. Previous studies have shown that the MAN in BD patients has increased FC with other networks, such as the dorsal attention network [42] and the DMN [43]. It is known that the MAN system mainly regulates sensory and motor functions. Abnormal FC related of MAN reflected the association with abnormal sensorimotor functions in BD patients. The MAN system in BD patients activate MAN more frequently, leading to abnormal sensorimotor function. Therefore, we hypothesize that the abnormal promiscuity of BD is related to brain sensorimotor functions.

### 4.3. Reduced Recruitment of BD Is Associated with Brain Cognitive and Memory Functions

In our experiment, the global recruitment coefficient of BD patients was significantly lower than the NC group. The recruitment coefficient of a brain region measures the probability that it belongs to the same community as the nodes in its subnetwork. It quantifies the intra-subnetwork module allegiance and reflects the communication of brain regions within the subnetwork. The decrease in the global recruitment coefficient implies that BD patients have fewer interactions within the network and less ability to communicate. We also found that at the subnetwork level, reduced recruitment coefficients were observed in the AN, reflecting reduced communication capability. Meanwhile, our study found significantly lower recruitment coefficients for bilateral middle frontal gyrus (MFG) and inferior frontal gyrus of the orbital part (ORBinf) regions at the nodal level. Recent studies have found that patients with BD have reduced FC in the MFG [44] and ORBinf [45] regions, which are important brain regions in the AN.

In addition, it has been shown that abnormalities of brain networks in BD patients are mainly found in brain networks related to external information processing, especially the AN [46], which affects cognitive functions. Furthermore, the MFG and ORBinf regions belonging to the AN are located in the prefrontal cortex. Abnormalities in the prefrontal cortex are a common manifestation of BD, which plays an important role in cognitive function [11,47,48,49]. Previous research has suggested that working memory deficits in BD patients may be due to abnormal prefrontal cortex activity [50]. Reduced prefrontal cortex activity implies reduced communication skills [51]. Our experiments found that reduced recruitment of the BD may contribute to cognitive and memory impairment from the perspective of dynamic reconfiguration of alliance preferences.

### 4.4. Reduced Integration of BD Is Associated with Brain Emotion Regulation Function

In the study, we found that the whole-brain integration coefficient is significantly reduced in BD patients. The integration coefficient of a region refers to the probability that it shares the same community with the nodes from other subnetworks. It quantifies inter-subnetwork module allegiance and reflects the communication of brain regions between subnetworks. The LSN showed a significant decrease in integration coefficients, reflecting inter-subnetwork communication abilities. Previous research has found that patients with BD have increased characteristic path lengths and reduced global efficiency [52], reflecting a disruption in global functional integration that leads to reduced communication capacity.

At the subnetwork level, we found that the integration coefficient of the LSN in BD patients was significantly reduced, reflecting its reduced ability to communicate with other subnetworks. Previous studies have found that the LSN in BD patients is biased towards greater isolation [53] and reduced communication with other regions. This is consistent with our results, suggesting that BD patients are more affected by the LSN, shifting from a state of equilibrium to a more isolated state. Moreover, the integration coefficients of the left hippocampus, right parahippocampal gyrus, bilateral amygdala, bilateral pallidum, and bilateral TPOmid regions were significantly reduced in LSN in our experiment. In addition, we further found that the decrease in the integration coefficient was most significant in the two groups of subnetworks, the LSN-AN and the LSN-DMN. Previous studies have shown that BD patients have impaired dynamic FC between the LSN-DMN and LSN-AN, which may be a biomarker for estimating suicide risk in BD patients [45]. This finding further complemented the greater impact of the LSN on the ability to communicate between subnetworks in BD patients.

In addition, we found that the integration coefficients of global and TPOmid.R region were significantly and positively correlated with HAMD and YMRS scores, indicating greater sensitivity to symptom severity. It indicated that the higher the integration coefficient, the higher the HAMD and YMRS scores. The integration coefficient represents the ability to communicate between subnetworks, while the HAMD and YMRS describe the level of depression and mania in BD patients, respectively. The higher integration coefficient implies overexcitation of the brain, which affects emotion regulation. It leads to more significant depressive and mania symptoms in BD. Notably, we found that 4/5 of the BD patients in this study had symptoms that were not severe, with YMRS and HAMD scores below 20. Therefore, our results may be applicable to mild BD patients. Whether the correlation results would be higher in severe BD patients remains to be explored. Our correlation results may provide evidence for abnormal integration of the BD is associated with brain emotion regulation function.

## 5. Limitations and Future Work

Several limitations of this study are worth considering. First, the scan time of our data was relatively short, and we obtained relatively little information about the dynamics, which failed to obtain the ideal results from the dynamic analyses. In addition, the sample size of our study is relatively small. To ensure the robustness and credibility of the findings, it would be worthwhile to repeat the experiment with a larger sample in the future.

## 6. Conclusions

In this study, based on a multilayer temporal network, the variation of inter-layer connections from different nodes over time was considered using inter-layer coupling. The dynamic reconfiguration of BD patients was also quantified by using the promiscuity, recruitment coefficient, and integration coefficient. We found that reduced inter-layer coupling strength and integration coefficient in the BD group were mainly caused by the LSN, increased promiscuity was mainly caused by the MAN, and reduced recruitment coefficient was closely related to the AN. These findings reflect the abnormal dynamic reconfiguration of BD patients and help us understand the pathological mechanism of BD from another perspective.

## Figures and Tables

**Figure 1 brainsci-14-00935-f001:**
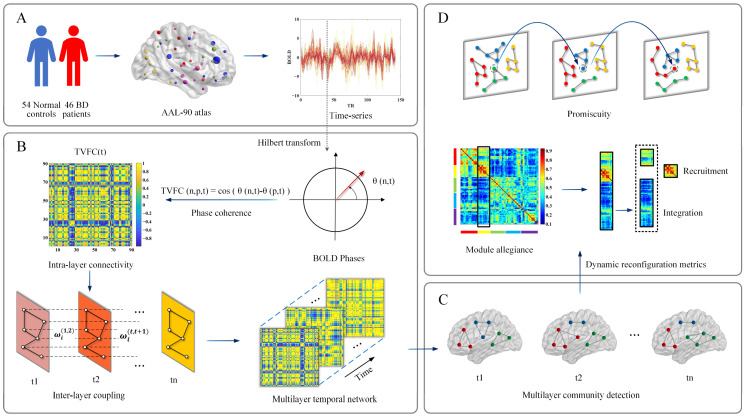
Flowchart of the methodology of this study. (**A**) Data preprocessing. Time series were extracted from the functional magnetic resonance imaging signal using AAL-90 atlas. (**B**) Multilayer temporal network construction. Hilbert transform was applied to the time series for each region. The functional connectivity of each time layer was estimated using phase coherence. The similarity is calculated to obtain the inter-layer coupling of the corresponding nodes of the neighboring time layers. (**C**) Multilayer community detection. Perform multilayer community detection for the constructed multilayer temporal network. Different colors indicate that the nodes are divided into different communities. (**D**) Dynamic reconfiguration metrics. Calculate the dynamic reconfiguration metrics recruitment coefficient, integration coefficient, and promiscuity (the node circled by dashed lines are involved in three modules) based on the results after dividing the communities.

**Figure 2 brainsci-14-00935-f002:**
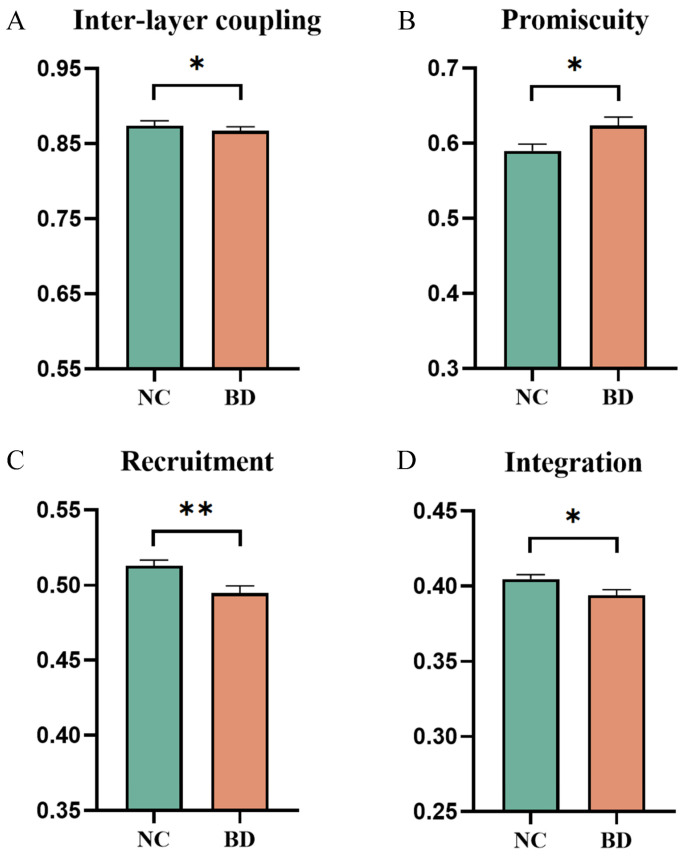
Group differences at the whole-brain level. (**A**) Inter-layer coupling strength. (**B**) Promiscuity. (**C**) Recruitment. (**D**) Integration. Asterisks indicate the *p*-values of significant group differences: * denotes *p* < 0.05 and ** denotes *p* < 0.01.

**Figure 3 brainsci-14-00935-f003:**
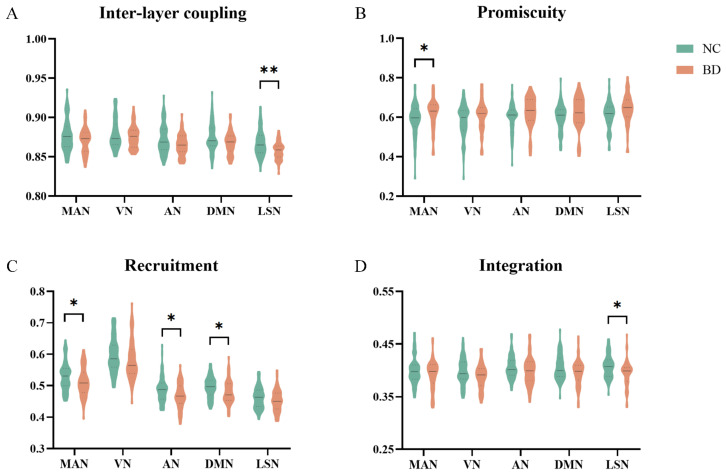
Group differences at the resting-state functional networks (RSNs) level. (**A**) Inter-layer coupling strength. (**B**) Promiscuity. (**C**) Recruitment. (**D**) Integration. Asterisks represent the difference between groups: * denotes *p* < 0.05 and ** denotes *p* < 0.01.

**Figure 4 brainsci-14-00935-f004:**
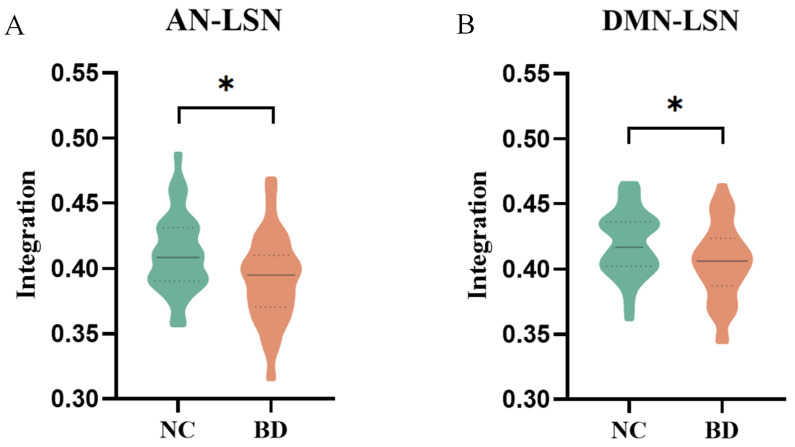
Group differences in integration for each pair of RSNs. Asterisks indicate the *p*-values of significant group differences: * denotes *p* < 0.05. (**A**) Group differences in integration for AN-LSN; (**B**) Group differences in integration for DMN-LSN.

**Figure 5 brainsci-14-00935-f005:**
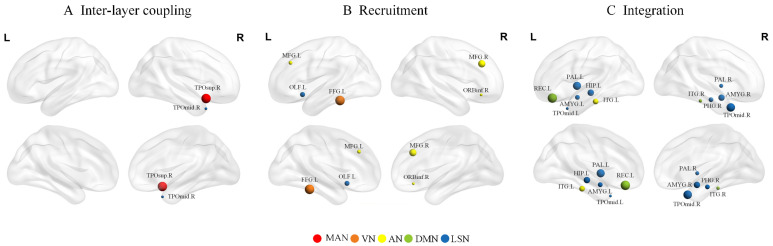
Brain maps of regions with significant differences in (**A**) inter-layer coupling, (**B**) recruitment and (**C**) integration. The size of the nodes is weighted by the *t*-value representing the difference between BD and NC, and the color represents the subnetwork to which the node belongs.

**Figure 6 brainsci-14-00935-f006:**
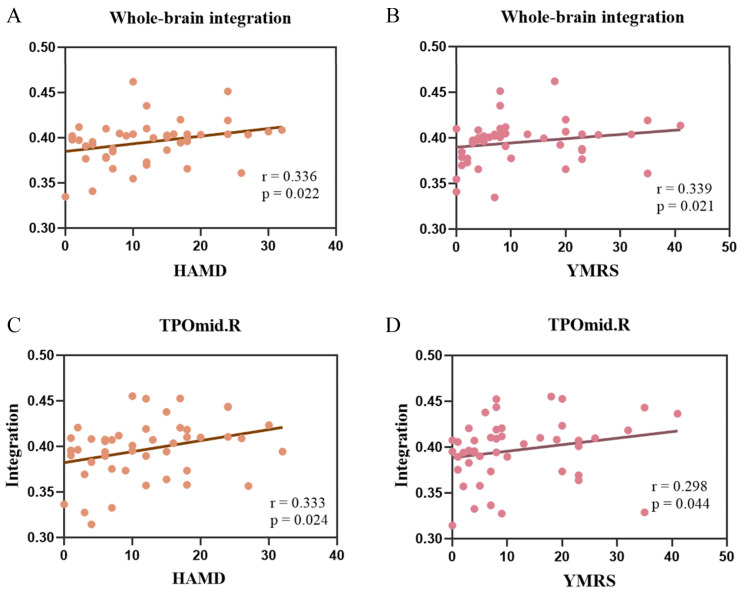
Spearman’s correlations between integration and Hamilton Depression Scale (HAMD) and Young Mania Rating Scale (YMRS) scores. (**A**) The correlation between whole-brain integration coefficient and HAMD score. (**B**) The correlation between whole-brain integration coefficient and YMRS score. (**C**) The correlation between integration in the right middle temporal gyrus of the temporal pole (TPOmid.R) region and HAMD score. (**D**) The correlation between integration in the TPOmid.R region and YMRS score. The dots in the figure represent each subject and the straight lines show the correlation between the integration coefficients and the scale scores.

**Table 1 brainsci-14-00935-t001:** Demographic and clinical characteristics.

Variables (Mean ± SD)	BD	NC	*p*-Value
Number of subjects	46	54	-
Age (years)	34.56 ± 8.75	34.57 ± 8.63	0.99
Gender (male/female)	26/20	30/24	0.92
Education level (years)	14.48 ± 1.86	14.91 ± 1.83	0.25
Duration of illness (months)	1.7 ± 4.32	-	-
Medication dose (mg/day)	774.45 ± 1054.82	-	-
HAMD	12 ± 8.43	-	-
YMRS	11.59 ± 10.62	-	-

Abbreviations: SD, standard deviation; NC, normal controls; BD, bipolar disorder; HAMD, Hamilton Depression Scale; YMRS, Young Mania Rating Scale.

## Data Availability

The data are publicly available at https://www.openfmri.org/ (accessed on 16 January 2023).

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
