# Peer review of "Abnormal Dynamic Reconfiguration of Multilayer Temporal Networks in Patients with Bipolar Disorder"

_brainsci, 2024, doi:10.3390/brainsci14090935_

Round 1

Reviewer 1 Report

Comments and Suggestions for Authors

The manuscript entitled "Abnormal dynamic reconfiguration of multilayer temporal net-2 works in patients with bipolar disorder” by Lai et al. constructed a multilayer temporal network and calculated the promiscuity, recruitment, and integration coefficients to compare dynamic reconfigurations between individuals with bipolar disorder (BD) and healthy controls (HC). The results indicated that BD had reduced temporal stability in attention and subcortical networks but increased promiscuity in somatosensory and auditory networks compared to HC. While the research topic is important and timely, I have multiple concerns that should be addressed prior to publication. These concerns are enumerated below:

  1. The introduction will benefit from a more in-depth review of the current literature investigating resting state functional connectivity in BD.
  2. Considering that a multilayer temporal network approach is not a conventional analysis method, the authors should add more details and examples for intra-layer and inter-layer connections as well as promiscuity, recruitment, and integration.
  3. The introduction is missing a justification for why intra-layer and inter-layer connections would be different between BD and HC.
  4. Although the authors use a secondary data analysis, they still have to report how the patients were diagnosed (M.I.N.I., SCID-5) and by whom. They also need to report inclusion/exclusion criteria, and clinical characteristics of the patients including their current mood state.  
  5. It is unclear why and how the AAL regions were separated into 5 networks. Please include a table for mapping ROIs on the networks. Also, please explain why such broad and non-specific networks were chosen instead of using Yeo’s or Smith’s architectures.
  6. The authors inconsistently report uncorrected and corrected p-values throughout the manuscript. Please add the table with uncorrected and corrected p-values for each analysis included in the manuscript.
  7. Figure 6 suggests that about 1/3 of BD patients could be in a manic episode at the time of the scan, which could increase head movement inside the scanner.  Considering that resting state fMRI is highly sensitive to head motion, it is important to discuss how head motion was related to the YMRS scores.
  8. It appears that more pronounced symptoms of mania and depression were associated with higher integration (Figure 6). However, HC had higher integration than BD (Figures 2-4), thus suggesting that more symptomatic patients were closer in connectivity metrics to HC than patients in remission. These findings are inconsistent with each other and warrant further discussion.
  9. Overall, the discussion lacks data interpretation from a clinical point of view and it remains unclear what promiscuity, recruitment, and integration characteristics tell us about brain function in bipolar disorder.

Reviewer 2 Report

Comments and Suggestions for Authors

In general, the study looks very interesting for investigation the neurophysiological causes of bipolar disorder. However, the manuscript omitted a description of a number of important details, without which the design and results of the study are difficult to understanding.

1. The most important terms for this study (multilayer temporal network, intra- and inter-layer coupling, network switching rates) are given without sufficient explanations. In the presented version of the manuscript, they are difficult for the understanding by a reader. My main difficulty concerns the meaning of the term "layer." Unfortunately, I could not understand what the authors mean by the word "layer" in the context of this article. I recommend giving definitions of what the authors mean by each term.

2. At the end of the Introduction section, I recommend formulating all the hypotheses that the authors test in their study. It is very significant that the authors explained whether they were going to test the hypothesis that BD has the different affects to the temporal dynamics of different brain networks.

3. It is necessary to describe in more detail the groups of participants. How exactly was the diagnosis of bipolar disorder, i.e. it was made only on the basis of questionnaires or also through an interview with a psychiatrist? Did patients have any comorbid disorders? How long have patients had BD? Have patients received any drug therapy?

4. The statistical analysis is not well described. The authors list a large number of metrics that they estimated based on fMRI analysis. Was the t-test applied to each of these metrics independently of the other measurements? How, then, did the authors assess the significance of the multifactorial effects? Multivariate comparisons cannot be made by t-test alone. For example, how were differences assessed for patient and control groups for different brain networks? It is also not clear to me how, using a t-test, you can assess the dynamic variability in connectivity.

5. I recommend redoing figure 2. In the presented version, the columns of the diagrams start from zero, which is not very convenient when viewing pictures. The error tails merge with the column boundaries. I recommend using 0.2 as the initial value of the diagram.

Round 2

Reviewer 1 Report

Comments and Suggestions for Authors

Thank you for addressing my comments.

Reviewer 2 Report

Comments and Suggestions for Authors

I recommend accepting the manuscript for publication in its current form.